# Dynamics of Endothelial Engagement and Filopodia Formation in Complex 3D Microscaffolds

**DOI:** 10.3390/ijms23052415

**Published:** 2022-02-22

**Authors:** Pierre Ucla, Xingming Ju, Melisa Demircioglu, Sarah Baiz, Laurent Muller, Stéphane Germain, Catherine Monnot, Vincent Semetey, Sylvie Coscoy

**Affiliations:** 1Institut Curie, Université PSL, Sorbonne Université, CNRS UMR168, Laboratoire Physico Chimie Curie, 75005 Paris, France; pierre.ucla@curie.fr (P.U.); xingming.ju@chimieparistech.psl.eu (X.J.); mdemircioglu@doctors.org.uk (M.D.); 2Chimie ParisTech, PSL University, CNRS, Institut de Recherche de Chimie Paris, 75005 Paris, France; 3CNRS, CNAM, PIMM, Arts et Metiers Institute of Technology, HESAM Université, 75013 Paris, France; sarah.baiz@ensam.eu; 4Center for Interdisciplinary Research in Biology (CIRB), College de France, Centre National de la Recherche Scientifique, Institut National de la Santé et de la Recherche Médicale (INSERM), Université PSL (Paris Sciences & Lettres), 75005 Paris, France; laurent.muller@college-de-france.fr (L.M.); stephane.germain@college-de-france.fr (S.G.); catherine.monnot@college-de-france.fr (C.M.)

**Keywords:** two-photon polymerization, microtopography, filopodia, endothelial cells, angiogenesis, contractility, mechanotransduction

## Abstract

The understanding of endothelium–extracellular matrix interactions during the initiation of new blood vessels is of great medical importance; however, the mechanobiological principles governing endothelial protrusive behaviours in 3D microtopographies remain imperfectly understood. In blood capillaries submitted to angiogenic factors (such as vascular endothelial growth factor, VEGF), endothelial cells can transiently transdifferentiate in filopodia-rich cells, named tip cells, from which angiogenesis processes are locally initiated. This protrusive state based on filopodia dynamics contrasts with the lamellipodia-based endothelial cell migration on 2D substrates. Using two-photon polymerization, we generated 3D microstructures triggering endothelial phenotypes evocative of tip cell behaviour. Hexagonal lattices on pillars (“open”), but not “closed” hexagonal lattices, induced engagement from the endothelial monolayer with the generation of numerous filopodia. The development of image analysis tools for filopodia tracking allowed to probe the influence of the microtopography (pore size, regular vs. elongated structures, role of the pillars) on orientations, engagement and filopodia dynamics, and to identify MLCK (myosin light-chain kinase) as a key player for filopodia-based protrusive mode. Importantly, these events occurred independently of VEGF treatment, suggesting that the observed phenotype was induced through microtopography. These microstructures are proposed as a model research tool for understanding endothelial cell behaviour in 3D fibrillary networks.

## 1. Introduction

The fundamental understanding of endothelium–extracellular matrix (ECM) interactions during the formation of new blood vessels meets considerable medical needs. The 3D geometrical organization of ECM fibres is complex, and modified in many pathophysiological conditions, such as in tumour microenvironments which favour the formation of tortuous and disorganized vessels. The global ECM stiffness, as well as local 3D fibre geometries, such as bundle size or alignment, are known to exert a central influence on cell fate. Indeed, the geometrical, chemical and mechanical properties of the microenvironment play a crucial role in defining cell behaviours, including morphology, function or differentiation [1,2,3]. In particular, the microtopographies of three-dimensional scaffolds surrounding cells, with subcellular sizes in the nanometre to micrometre ranges, have been shown to elicit responses such as contact guidance [4,5,6,7,8,9,10], the modulation of migration or invasion [3,4,11,12,13] or the control of cell fate and differentiation [14,15,16]. In the field of bioengineering, a comprehensive range of methods has been developed, either to mimic the native microenvironment, or to create innovative microtopographies eliciting cell responses of medical interest. These techniques include photolithography [17], colloidal templating [14], electrospinning [18], moulding [19,20], 3D impression [21] or two-photon polymerization (TPP) [22,23,24]. To date, TPP allows the largest precision for the creation of 3D microstructures, with typical resolutions of 0.2 µm in the *xy* planes and 1 μm in the *z* plane, making possible the creation of complex microstructures guiding precise migration and invasion [25], or with coupled measurements of 3D mechanical properties [24,26]. Notably, the culture of endothelial cells on microstructures such as microgrooves or micropillars modulates their phenotype and their migratory properties [27,28,29]. 

Controlling the sequential steps involved in angiogenesis meets considerable medical needs. Indeed, angiogenesis is deleterious in tumour progression and metastatic dissemination, but beneficial in chronic ischemic diseases, and desired for the vascularisation of bioimplants. The technologies currently developed to trigger angiogenesis rely on growth factors released from bioengineered matrix or cells [30,31] or on an in vitro approach based on spatial patterning with Dll4 ligand [32]. The specific role of microtopography has been poorly characterized. The influence of microgratings and micropillars on endothelial cells was recently reported, using hierarchical nano- and micro-gratings, enhancing the capacity to drive in vitro 3D angiogenesis after cell dissociation from microstructures [33]. However, more complex 3D structures are required in order to trigger specific 3D events characteristic of the initiation of angiogenesis. Early steps in the angiogenic process imply a transitory specialization of some cells into tip cells, which exhibit filopodia [34] involved in the migration and exploration of the microenvironment, and generating extended membrane protrusions named dactylopodia central for tip cell invasion [35]. These cells invade the extracellular matrix, dragging other endothelial cells (stalk cells) with lumen formation, and the balance between tip and stalk cells is ensured by a negative Notch/Dll4 feedback loop; finally, fusion between tip cells emanating from different capillaries gives rise to new vessels (anastomosis) [36]. These angiogenic mechanisms are triggered by chemical gradients of growth factors, in particular VEGF (vascular endothelial growth factor), and by the dynamic interactions with the extracellular microenvironment. It is still unclear which phenotypic characteristics of tip cells are triggered by chemical cues (VEGF) or can be induced by microtopographical cues (ECM geometry).

Here, we describe in detail the interaction between an endothelial monolayer and two types of underlying 3D patterns, «closed» hexagonal lattices and «open» hexagonal lattices on pillars. We observe that, while endothelial monolayers stay at the surface of «closed» hexagonal lattices, their culture on «open» hexagonal lattices on pillars leads to endothelial engagement from the monolayer and to the generation of filopodia, independently of exogenous growth factors (VEGF). This suggests that these behaviours are induced by microtopography. In particular, the protrusive state observed in open structures, based on filopodia dynamics, contrasts with the endothelial cell migratory state on 2D substrates, exclusively based on lamellipodia. Thanks to the development of image quantification tools, we probe the influence of the microtopography (mesh size, regular vs. elongated structures, role of the pillars) on the orientations and engagement of endothelial cells and on filopodia dynamics, and we characterize the mechanisms involved in the interaction of endothelial cells with these 3D substrates, leading to the generation of a phenotype reminiscent of tip cell organization.

## 2. Results

### 2.1. Fabrication of Hexagonal Microstructures and Cell Seeding

We investigated the role of complex microtopographies in the control of endothelial protrusive behaviour by using 3D microstructures generated by the two-photon polymerization of NOA resin (Norland Optical Adhesive). We studied cell behaviours in 3D geometries derived from hexagonal patterns. We previously described simple 3D hexagonal lattices for the generation of deep epithelial protrusions strongly increasing cell basal surface area [37], and we were interested in testing the behaviour of endothelial cells on these scaffolds, referred to here as «closed» structures (Figure 1a). However, the physiological microenvironment of endothelial cells is not compact, but composed of extracellular matrix fibres, whose fibrillary characteristics play a key role in their adhesive behaviour, and in particular in guidance events during angiogenesis. So, we performed a systematic comparison between closed microstructures and derived hexagonal lattices on vertical pillars of similar dimensions, which allowed the physical communication of cells in the bottom plane (“open” structures, Figure 1b,c). We focused on 7 µm-high structures, including a typical pillar height of 2–4 µm (Figure 1b,c,f,g). Hexagons were either regular (Figure 1b) or elongated (Figure 1c). Elongated microstructures were considered in the aim to obtain elongated cell shapes, as expected for endothelial cells in capillaries. Our rationale to use elongated structures was to observe if protrusive migratory events were facilitated in this context: indeed, endothelial cells were shown to colonize substrates with grooves faster than flat substrates [27]. We adopted the following nomenclature: regular hexagonal lattices with a horizontal *D* dimension, e.g., of 8 µm, were referred as *D8-open* or *D8-closed* structures, while *l7L14-open* or *l7L14–closed* referred to open or closed structures with elongated hexagons of 7 µm × 14 µm. We built closed structures (Figure 1e) and open structures mechanically stabilized with an external crown of closed hexagons (Figure 1f,g).

HUVECs were seeded on microstructures and fixed after cell coverage of the structures (generally 1–3 days). Unless otherwise specified, a standard VEGF concentration (0.5 ng/mL) of endothelial growth medium was used. We characterized cell coverage on top of the microstructures (Section 2.2) and the topographical requirements for cell engagement into the structures (Section 2.3), and focused in particular on mechanisms for the microtopography-driven formation of endothelial filopodia (Section 2.4, Section 2.5 and Section 2.6).

**Figure 1 ijms-23-02415-f001:**
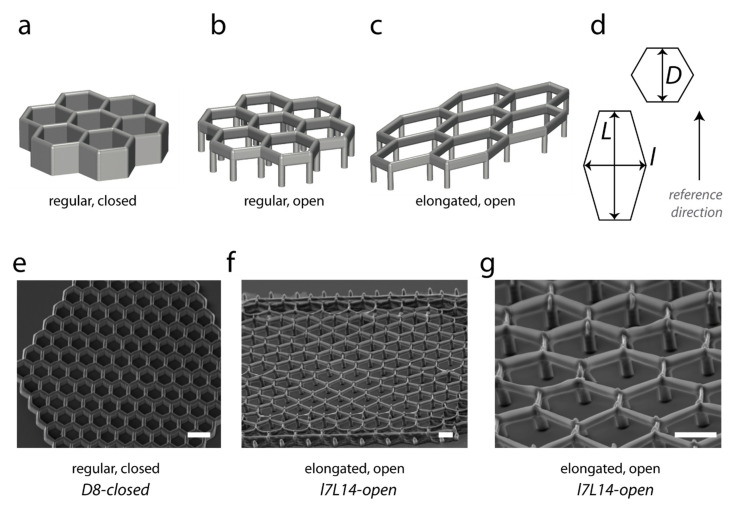
Microfabrication of microstructures. Microstructures were realized with NOA61 (Norland Optical Adhesive) by two-photon polymerization. (**a**) Hexagonal lattices previously described [37] are referred to here as “closed” microstructures. (**b**,**c**) “Open” microstructures were realized by building hexagons on pillars, with a total height H of 7 µm (including the typical pillar height of 2–4 µm), and variable horizontal dimensions. Regular (**b**) and elongated (**c**) hexagons were used. (**d**) Scheme of dimensions measured for regular (D) and elongated (l, L) hexagons. (**e**–**g**) SEM imaging of *D8-closed* (**e**) and *l7L14-open* (**f**,**g**) structures. (**e**) Top view, (**f**,**g**) side views, bars 10 µm.

### 2.2. Formation of an Endothelial Top Monolayer on Top of the Structures, and its Orientation in the Function of the Geometry of the Underlying Microstructure

A few days after cell seeding, endothelial cells efficiently covered the different microstructures (Figure 2a–d; see Figure A3a for early coverage), and organized in monolayers on top of the structures, in a similar manner on either closed or open. The orientation of cells on the top monolayer, as assessed by nuclei orientation, was compared on elongated hexagons (Figure 2a,c) and on regular hexagons (Figure 2b,d). After the tridimensional segmentation of the nuclei of the top plane (Figure A1a), the angle between the principal nucleus axis in the horizontal plane and the reference axis was determined (Figure 2a,c, vertical arrows). The nuclei in the top plane were oriented in the direction of hexagonal elongation on elongated *l7L14* microstructures, both closed or open; mean angles with the reference direction were, respectively, 30 ± 21°, *n* = 127, and 29 ± 24°, *n* = 700) (Figure 2e,f). This preferential orientation appeared to not be affected by the addition or removal of VEGF (Figure A1b,c). On the contrary, nuclei orientation was random on regular *D4.5*, *D6.6*, *D8.8*, *D10*, or *D13.5-open* microstructures (44 ± 26°, *n* = 1051, with a mean of 45° expected for a random orientation, Figure 2g; see separate similar behaviours in Figure A1d–h). The elongation of cell shape paralleled nuclei orientation from visual observations (Figure 2c, Figure 3 and Figure A3b), as expected from the literature [38,39,40]. 

Thus, elongated hexagonal geometries induced an elongation of HUVECs nuclei guided by the main direction of hexagons, as expected from the literature on endothelial cells on grooves. Some papers also reported an increased colonization rate for endothelial cells on grooves or elongated structures [27], which could be considered for future bioimplant applications. 

### 2.3. Endothelial Engagement from the Top Monolayer in the “Open” Microstructures Only

#### 2.3.1. Global Observations on Endothelial Cell Engagement

During the first angiogenic step, tip cells polarize, sprout out of the endothelial monolayer and engage in the surrounding microenvironment. In our closed microstructures, there was no cellular vertical engagement from the top monolayer (Figure 3a). The cell cortex was not able to extend further than 1–2 µm below the top of the structure, except in rare occurrences (only 0.6% hexagons filled, and 3% partly filled, on 1105 closed hexagons, with occasionally a few vertical filopodia (see Appendix A). On the contrary, cells were able to engage vertically in open structures, and to come into contact with the bottom substrate 7 µm underneath, while still maintaining the top monolayer (Figure 3b–e, see Appendix A). The formation of F-actin-rich vertical protrusions could occur without nucleus engagement, likely corresponding to the early steps of colonization (Figure 3b). However, in most cases a deep nucleus engagement towards the bottom substrate was observed, with some nuclei undergoing strong deformation due to the micropillars (Figure 3c–e). Another striking feature in the open configuration was the formation of endothelial filopodia emerging from the vertically engaged cell body in contact with the bottom substrate (Figure 3b–e). Filopodia were already present at early steps of engagement, where they could either arise directly from cell membrane around pillars (Figure 3b), or in later stages; in particular, they were observed in the extension of elongated protrusions reminiscent of the recently described endothelial dactylopodia (Figure 3d). It is noteworthy that a frequent enrichment in F-actin around pillars was observed (Figure 3e, white arrow). Membrane winding around pillars may provide anchor points to cells beginning to engage vertically, as well as favour focal adhesion maturation and filopodia formation. In addition to this mesenchymal protrusive mode (filopodia + dactylopodia-like), an amoeboid organization with membrane blebs was occasionally observed in open structures; transitions between the two modes will be discussed later (see Section 2.5).

#### 2.3.2. Quantification of Nuclei Engagement in Function of the Microtopography

Endothelial vertical engagement was further quantified by measuring the penetration of nuclei into the structure. After 3D nuclei segmentation, the bottom *z* position of each nuclei was recorded (Figure 4). Again, the ability to engage nuclei from the top monolayer was dependent on the underlying geometry, closed or open (Figure 4a,b). In closed structures, most nuclei remained on top of the structures or ~2 µm above (*l7L14–closed*, only 2% of nuclei below 5 µm, Figure 4a), even after the addition of VEGF 5 ng/mL (Figure A1i). In contrast, open structures with the same horizontal dimensions induced significant vertical nuclei engagement up to the bottom of the structures (*l7L14–open*, 62% of nuclei below 5 µm, Figure 4b). The same engagement was observed in open structures even without VEGF (Figure A1j).

The vertical engagement is also expected to depend on the horizontal mesh size [37]. Therefore, we studied the engagement of HUVECs on open regular structures of different dimensions D, from 13.5 µm to 4.5 µm (Figure 4c–g). As for *l7L14-open* structures, the majority of nuclei were able to engage vertically and to contact the bottom substrate for *D13.5-open*, *D10-open* and *D8.8-open* structures (Figure 4c–e), in agreement with the fact that *D8.8-10* structures have a surface area close to *l7L14* structures (respectively, 58%, 78% and 64% of nuclei below 5 µm). Nuclei engagement was intermediate for *D6.6-open* structures (44% nuclei below 5 µm, Figure 4f). In contrast, almost no engagement was observed in smaller *D4.5-open* microstructures (6% nuclei below 5 µm, Figure 4g).

To summarize, HUVECs formed a monolayer in the top of every microstructure studied, but was able to engage nuclei vertically from this monolayer only for large enough and open structures (Figure 4h). So, the behaviour of HUVECs was in contrast with that reported for epithelial cells, which were able to engage massive vertical protrusions from dense monolayers on closed hexagonal lattices, as described in our previous work [37].

### 2.4. Induction of Endothelial Filopodia in Open Microstructures: Development of Automated Detection and Tracking, and Filopodia Characteristics

Filopodia were observed consistently in open structures, emerging horizontally from vertical protrusions in contact with the glass substrate, as visualized from F-actin and membrane labelling (Figure 3b–e and Figure A3c). This contrasted with the lamellipodia-based endothelial organization observed on sparse or confluent 2D substrates, where filopodia have not been observed [35]. Indeed, 2D monolayers of confluent endothelial cells display either continuous, straight and stable or irregular and serrated cell–cell junctions when mature or immature, respectively [41], without displaying protrusive filopodia, and are instead characterized by the formation of large adhesive lamellipodia involved in the migration process [42]. We focused in more detail on filopodia characteristics and dynamics in order to answer the following questions: (1) do filopodia have a protrusive, exploratory behaviour (in opposition to filopodia-like structures that would come from cell retraction [43])? (2) How do their characteristics compare with values found for filopodia in general, and for in vivo endothelial filopodia in particular?

Filopodia characteristics in the standard growth conditions (0.5 ng/mL VEGF) generated in microstructures were quantified using home-made software, with technical challenges being the intricated multicellular organization and the presence of the autofluorescent structure pillars (see Appendix B
Figure A2, Figure 5 and Figure 6). First, a semi-automated home-made Matlab script was developed to detect and quantify filopodia on fixed images (Figure A2 and Figure 6a,b). Second, segmented images obtained in the first steps were used to train a convolutional neural network, to perform automated filopodia detection and tracking on timelapse images of HUVECs Lifeact-GFP, in order to quantify filopodia lengths, numbers, orientations, formation/retraction rates, and the percentage of elongating filopodia (Figure 5 and Figure 6c–g, Table 1).

The length of filopodia could extend to 10–15 µm, with an average length of 4.18 ± 2.97 µm (median 3,43 µm, *n* = 1651 filopodia, in 21 structures) as assessed from fixed samples (Figure 6a,b). Filopodia orientations compared with structure axes were computed; indeed, the geometry of pillars may be important for filopodia formation, in line with the observation that a significant number of filopodia emerged near pillars and extended preferentially in the direction of other pillars (Figure 3b–e). Indeed, we observed an asymmetry of filopodia orientations in elongated *l7L14-open* hexagonal lattices, due to the asymmetry of pillar organization, in particular for long filopodia (lengths > 5 µm) (Figure A4), while in regular microstructures filopodia had globally isotropic orientations (Figure A5g,h).

At last, time-lapse imaging allowed the extraction of dynamic values from numerous filopodia (typically, 10–100 filopodia were detected at each time point in a movie, and several thousands in total during the observation time). Lengths, normalized numbers, filopodia lifetimes and rates of formation and retraction of filopodia (−) and (+) extremities were extracted from time-lapse images, and values averaged for each independent movie are shown in Figure 6c–g and Table 1. The mean length value obtained from timelapse samples was 3.36 ± 0.41 µm, the mean lifetime ~230 s, and elongation and retraction rates were of the order of 40–50 nm/s. These values are in the range of typical values described in the literature for filopodia dynamics [35,43,44], in particular filopodia extension rates of 55–87 nm/s reported in different systems [43]. It is noteworthy that filopodia characteristics, such as length, number or dynamic behaviour, exhibited no clear difference between elongated and regular structures (except for filopodia orientations).

### 2.5. Molecular Mechanisms Governing the Transitions between the Different Protrusive Modes Characteristic for the Microstructures

In addition to filopodia quantification, we studied the interdependence of the different protrusive forms present in the microstructures. First, we observed switches between a mesenchymal phenotype (adherent protrusions) and an amoeboid one (membrane blebs). Second, within the mesenchymal mode, two types of protrusions were seen, filopodia, connected or not to dactylopodia-like extended protrusions; from the literature, dactylopodia-like protrusions may derive from filopodia in a controlled way and may play a central role during invasion and migration in non-vascular ECM [35]. These two types of switches between modes are here addressed sequentially.

#### 2.5.1. Reversible Switches between Mesenchymal (Filopodia + Dactylopodia-Like) and Amoeboid Modes Are Observed, with ROCK and MLCK Activation of Actomyosin Contractility Favouring Distinct Migration Modes

Spontaneous mesenchymal–bleb transitions

Movies on Lifeact-GFP cells in the microstructures revealed the existence of transitory membrane blebs in parts of the structures, while other cells in the microstructures retained their adherent protrusive phenotype, and cells outside the structures remained in a lamellipodia-based mode. In the microstructures, interconversions could occur spontaneously between the adherent (mesenchymal-type) mode and the blebbing mode (Figure 7a,b and Appendix A). Note that blebs were also described in vivo and may contribute to plasticity in sprouting angiogenesis [45]. Since the major difference with cells outside the structures was cell confinement, we assume that reversible membrane blebbing may derive from it [46], with the possible involvement of nuclear confinement during migration between pillars (see Discussion).

Mesenchymal–bleb reversible transitions in function of ROCK/MLCK balance

In order to understand the molecular players involved in the transition between blebs and adherent states, we targeted ROCK and MLCK pathways. Indeed, in the literature the transition between lamellipodia and membrane blebs was expected to be governed by the degree of non-muscle myosin II activity, with ROCK favouring a blebbing phenotype and MLCK a mesenchymal adherent phenotype [47]. Different contractility levels (higher with ROCK than with MLCK) may be at play [47], as well as a difference in spatial distribution, ROCK activity being higher in the centre of cells and MLCK in their periphery [48].

Inhibiting MLCK with ML-7 (10 µM) induced filopodia retraction and bleb formation into the microstructures, but not outside (Figure 7c and Figure A7a, see Appendix A). Almost all cell islands exhibited an amoeboid behaviour 30 min after ML-7 addition (on *n* = 3 movies). This was in agreement with a strong decrease in filopodia number observed after quantification (Figure 8b and Figure A7b). The other dynamic parameters of the few remaining filopodia, lengths, rates of elongation and retraction, and lifetime, were not affected (Figure 8a,c and Figure A7b,f). Specific effects of MLCK inhibition on filopodia stability were reported previously, with ML-7 inhibition experiments suggesting that acto-myosin activity was required for the maturation of filopodia shaft adhesions in fibroblasts [49].

On the contrary, ROCK inhibition by Y-27632 (10 µM) inhibited bleb formation (Figure 7d, see Appendix A), in agreement with [47] and as already observed for endothelial cells [50]. All membrane blebs present before drug addition shifted to an adherent state after Y-27632 treatment (from 5 to 0 blebbing cell islets in three movies). Numerous protrusive filopodia were still observed, and movies visually suggested the formation of a more branched filopodia network upon Y-27632 addition (which would be in agreement with protrusive events linked to LIM kinase 1 (LIMK-1)-mediated phosphorylation of the actin-depolymerizing factor cofilin in that context [51]). Nevertheless, from quantitative data, the dynamic filopodia parameters studied did not significantly differ after Y-27632 treatment (Figure 8a–c and Figure A7c,g). Interestingly, it was also observed in growth cones that Rho kinase inhibition had much smaller effects on growth rate and filopodia numbers than MLCK inhibition [52].

#### 2.5.2. Dactylopodia-Like Protrusions and Their Stabilization by FAK Inhibition

Dactylopodia are finger-like, extended endothelial protrusions proposed to be central for non-vascular ECM invasion. They originate from endothelial filopodia through Arp2/3-dependent membrane ruffling at the base of filopodia. The balance between dactylopodia and filopodia is controlled by NMIIA myosin, which promotes the maturation of focal adhesions (FA), thereby limiting Arp2/3 activation in nascent adhesions [35]. Extended protrusions evocative of dactylopodia were frequently observed in our microstructures and were connected to filopodia. The sizes observed for these protrusions in the microstructures were similar to the ones reported in the literature: typical lengths observed were 25 µm in our system (8–57 µm), for a mean width of 2.8 µm (1–6) (*n* = 47 dactylopodia-like protrusions on 12 movies), to be compared with reported lengths of 20 µm (5–33) and width 2 µm (0.7–5) found in vivo [35] (Figure 3d). Movies revealed that dactylopodia-like protrusions were derived from filopodia, with the same triggering mechanism of membrane ruffling at the base of filopodia as the one described in the literature [35] (see Appendix A). In order to study if their formation from filopodia was favoured in the context of non-mature FA, as would be expected, we used the FAK inhibitor PF-573228. We observed that PF-573228 (10 µM) indeed stabilized dactylopodia-like protrusions (Figure 7e, see Appendix A), with a ~3-fold increase in the number of dactylopodia-like protrusions after treatment (from 8 to 24 dactylopodia-like protrusions in three movies). The quantitative analysis suggested an increase in filopodia lifetime (Figure 8c and Figure A7d), without a major effect on the other dynamic filopodia parameters (Figure 8a,b and Figure A7d,h). Note that PF-573228 also induced an amoeboid–mesenchyme transition by reducing (about 2-fold) the number of blebbing cell islets, in agreement with the fact that FAK inhibition regulates endothelial membrane blebbing by reducing actomyosin contractility [50].

Then, dactylopodia-like protrusions in our microstructures share common characteristics, not only morphologically but also mechanistically, with dactylopodia that were described as central for endothelial invasion.

### 2.6. Independence on VEGF

In sprouting angiogenesis, filopodia and dactylopodia formation are induced by VEGF signalling, acting on VEGFR2/3-NRP1, and triggering the CDC42 signalling pathway, regulating formins and thus filopodia formation, and the SRF (Serum Response Factor)/MyosinII pathway, modulating contractility [35]. However, we observed vertical elongation and filopodia formation in the low VEGF concentration present in the basal HUVEC culture media (VEGF 0.5 ng/mL: Figure 3). Such a low concentration is unable to trigger angiogenesis in classical 3D angiogenesis assays [53]. We therefore analysed the VEGF dependence of filopodia formation (Figure 9). We first checked that VEGF was efficiently delivered to subcellular parts in the bottom of the structures, by labelling phosphorylated VEGFR2 after a short VEGF stimulation (Figure A8). We then performed cultures with or without continuous VEGF stimulation. We tested two different culture media without VEGF (Figure 9a,b), standard VEGF concentration (Figure 9c), and higher VEGF concentrations (5, 20 and 50 ng/mL, Figure 9d–f). In classical 3D assays, 5 ng/mL VEGF was sufficient to trigger angiogenesis, and a saturation of the system occurred with 50 ng/mL VEGF [54]. We observed vertical engagement and filopodia formation in all conditions, with similar length and density characteristics (Figure 9g–h). These striking results show that endothelial filopodia are formed independently of exogenous VEGF in our system.

To exclude that an endogenous VEGF secretion by HUVECs may play a role in the observed protrusive phenotype, we performed short treatments with sunitinib, an RTK (receptor tyrosine kinase) inhibitor, in living cell experiments. Sunitinib (300 nM) did not significantly affect the presence of filopodia or dactylopodia-like protrusions, nor did any parameter analysed for filopodia dynamics (Figure 9i,j and Figure A7e, see Appendix A). Here, we mainly used short treatments with sunitinib targeting the quick cell responses [55,56], which involves the cytoskeleton organization and its upstream signalling pathways. We also checked that numerous protrusive filopodia were still present after longer sunitinib treatments (5 h and 18 h, Figure 9i and Appendix A

Therefore, the observed filopodia and filopodia-derived protrusions are likely induced by microtopographies.

## 3. Discussion

### 3.1. Summary

In this paper, we report that endothelial cells organize in monolayers at the surface of NOA hexagonal lattices. In open microstructures (hexagonal lattices on pillars), cells were able to engage vertically from the monolayer and to generate endothelial filopodia, two events evocative of a tip-like phenotype. Two-photon polymerization allows a precise 3D control of microstructures, with here an interplay between horizontal elongation and vertical pillars in order to control the endothelial phenotypes, as illustrated in the recapitulative Figure 10, top: (1) On top of the microstructures, the elongation of cells in the monolayer is controlled by the horizontal elongation of hexagons. (2) Endothelial cells are able or not to engage from the monolayer depending on the presence of vertical pillars. (3) In open microstructures, horizontal filopodia are induced as cells engage vertically, with filopodia orientation partly dependent on pillar geometry and thus on horizontal hexagonal elongation.

In open microstructures, we observed interconversions between different protrusive modes characteristic for endothelial organization in 3D substrates (Figure 10, bottom). First, an amoeboid mode, probably due to confinement constraints, was occasionally observed. Second and more systematically, a filopodia-based organization was observed, from which could originate dactylopodia-like elongated protrusions (mesenchymal modes). Dactylopodia were reported to play a central role in non-vascular ECM endothelial invasion [35]. Importantly, filopodia and derived protrusions were observed independently on VEGF, suggesting they may be induced by geometry.

We will now address in more detail the following points: possible mechanisms for vertical engagement, the influence of confinement, the mechanisms for filopodia and dactylopodia-like induction, signalling pathways, and potential applications that may derive from this study.

### 3.2. Vertical Engagement Inside the Microstructures

We found that endothelial cells could only engage vertically from the top monolayer in open microstructures (contrary to epithelial cells, which could form deep basal protrusions even in closed structures [37]). While the precise determination of the mechanisms involved is beyond the scope of this study, we propose a putative mechanism where cells initiating an engagement in vertical open columns would be stabilized by anchor points around pillars in the bottom part, as suggested by the frequent intense F-actin rings (see Figure 3e). Such a mechanism would be in good agreement with the behaviour observed for endothelial cells on micron-sized fibres (such as our pillars), which could form protrusions winding around fibres [57,58]. The formation and 1D migration of protrusions around such fibres was reported for a variety of cell types including endothelial cells [57,58], with different possible regimes and possible rotations around fibres [57,58,59,60]. Cells adopted spindle-shaped morphologies, and long protrusions along the fibre could span up to several hundreds of µm in length and exhibited protrusive waves governed by actomyosin dynamics, according to the balance between Rac1-Arp2/3 and Rho-formins pathways [57]. The protrusive cell activity around 1D fibres has largely been proposed to mimic the 3D ECM microenvironment. It was reported in human carcinomas that an increased ECM alignment provided tracks leading to endothelial activation and capillary morphogenesis, associated with the increased endothelial expression of angiogenesis-related markers and of α1-α3 integrins [61].

### 3.3. Confinement and the Amoeboid Protrusive Mode

Since the amoeboid mode was mostly observed into the microstructures, and not outside of them, we hypothesized a role for cell confinement. A plausible mechanism was the involvement of nuclear confinement, since cells were proposed to up-regulate their actomyosin contractility in response to confinement via nuclear envelope stretch-sensitive proteins [46]. It is important to note that this mode of blebbing was reported as nonapoptotic [46], and is indeed reversible in our experiments. In our microstructures, cells were subjected to two main types of confinement: the first when engaging into the microstructures in the hexagons; the second only occasionally, depending on the way cells migrated between pillars. The first engagement in hexagons of 14 µm × 7 µm was unlikely to trigger the blebbing phenotype alone, first because in that case blebs would systematically be observed in the structure, and second because in the literature the upregulation of contractility was observed when cell height was restricted to 5 µm, but not 10 µm. So, we hypothesize that the protrusive mode in the microstructures depends partly on the extent of nuclei confinement met at a particular time during cell migration between pillars. Blebs reported in vivo were proposed to contribute to the plasticity in sprouting angiogenesis [45], or could also be caused by confinement events in the ECM.

### 3.4. Filopodia Generation and Induction of Dactylopodia-Like Protrusions

We found that our microstructures allowed endothelial cells to switch from a 2D lamellipodia-based dynamic behaviour to a filopodia-based morphology characteristic of 3D migration and angiogenesis onset. The role of filopodia in the detection of micro- and nanotopographies has been widely documented for different cell types [62,63,64,65,66]. In particular, endothelial filopodia generated by nanotopography were reported for bacterial pili, silk fibroin films, anodic alumina with surface functionalization and anodized titanium dioxide [62,63,64,65,67]. Filopodia induction generally involved guidance along nanotopographical cues, following different passive or active mechanisms. Filopodia originated from 1D membrane wetting in the case of bacterial pili, without active generation by actin polymerization [62,68]. In other studies, cytoskeleton remodelling or differentiation events favouring filopodia formation and induced by the nanotexturation were reported: the activation of VEGFR2 and eNOS (endothelial nitric oxide synthase) [65]; and the upregulation and clustering of αvβ3 or α5β3 integrins, resulting in activity of focal adhesions (FAK phosphorylation) and in the activation of the PI3K/Akt pathway involved in angiogenesis [65,67]. This feature of guidance by nanostructures was not restricted to endothelial cells, but was also described for bone cells [69,70,71,72]. These behaviours contrast with the generation of horizontal filopodia in our system, which were not guided along structures; indeed, aside from micron-sized vertical pillars, we do not expect a nanostructuration of our bottom glass substrate, and never observed it in electron microscopy. Filopodia dynamics in our system corresponded to protrusive events, characteristic of an active exploratory behaviour. The detailed dynamic analysis in our system allowed us to quantify filopodia features, which proved to be remarkably stable under the different conditions, with two exceptions, probably pointing to the robustness of their regulation.

We observed that filopodia formation involved transitory contacts with pillars. Mechanisms of stabilization on pillars were reported for fibroblasts near highly flexible hairy silicon nanowires, allowing us to obtain measurements on traction forces exerted by filopodia (in the range of nN) [66]. Although on a different cell type (fibroblasts spontaneously generate many filopodia, contrary to endothelial cells), it is of interest to note that in this study, filopodia exhibited a clear topographical preference for pillar bundles over a flat substrate, with most filopodia tips in contact with flexible pillars, and unstable filopodia on the flat substrate [66]. In our microstructures, we also distinguished two types of filopodia, with longer ones that had a preferential orientation in the principal direction of hexagons in elongated microstructures. The tips of longer filopodia were usually not in contact with a pillar; rather, pillars were tangential to filopodia. This suggests an intermediate stabilization by pillars, giving a general orientation to filopodia during active exploratory events.

What is the physiological importance of generating endothelial filopodia? While filopodia formation is characteristic for tip cells, in vivo filopodia were described as dispensable for the induction of tip cells and for cell guidance, but modulated the migration speed of the endothelial tube in formation, and were required for anastomosis [73]. More recently, the importance of filopodia for sprouting angiogenesis was proposed to involve their ability to generate the extended dactylopodia protrusions required for endothelial invasion [35]. Indeed, such protrusions were observed in our microstructures, and we observed that they were derived from filopodia from our living experiments, together with the quantification of the dynamic filopodia parameters modulated during this transition. So, the microtopographies presented here allow the generation of filopodia and derived dactylopodia-like protrusions, both expected to play a role in the angiogenic process.

### 3.5. Signalling Pathways

Although the protrusive phenotypes induced by our microstructures—vertical engagement from a monolayer and the generation of endothelial filopodia—are characteristic for tip cells, the signalling pathways involved differ from the physiological pathway at work in angiogenesis. The canonical pathway for tip cell activation is VEGFR2 activation by VEGF, which triggers several pathways, including PI3K/Akt, leading to cytoskeleton remodelling and filopodia formation; CDC42 and Rac1, which govern endothelial membrane protrusions; and FAK, involved in junction maturation. A fundamental observation of our study is that vertical engagement and the formation of filopodia and derived dactylopodia-like protrusions occur efficiently without exogenous VEGF, or upon RTK inhibition. This suggests the involvement of pathways alternative to VEGFR2 activation. Our preliminary experiments did not allow us to identify an involvement of the known alternative pathway NRP1 (neuropilin 1), a VEGFR2 co-receptor downstream of VEGF signalling, that also participates in ECM-dependent CDC42 recruitment, generating filopodia [74,75], or of the YAP/TAZ pathway [76]. Micro- and nanostructurations are known to modulate the state of maturation of FA [77], which could partly intersect with Src/FAK signalling triggered by VEGF stimulation. It will be of key importance to establish if other characteristics of angiogenic activation will occur in derived microstructures, in the same way as previously reported for nano- and micro-gratings, enhancing the capacity to drive in vitro 3D angiogenesis after cell dissociation [33].

### 3.6. Perspectives

We describe here hexagonal microstructures promoting protrusive phenotypes characteristic for endothelial tip cells, but without VEGF gradients. On a fundamental point of view, because two-photon polymerization is a versatile way to play on the geometry but also on the local chemistry, including adhesion and stiffness, the system described here provides a convenient way to decipher endothelial filopodia sensing, setting cell migratory properties in 3D fibrillary ECM networks [78]. Furthermore, this study opens paths towards two main applications. First, these open microstructures could be a promising tool for angiogenesis induction and vascularization in the field of bioimplants. Indeed, the success of tissue engineering relies on a rapid and efficient blood supply, therefore requiring the controlled generation of a capillary network inside the implant and its anastomosis within the host vasculature. While some approaches consist of providing the local production of angiogenic growth factors by co-culturing in the implant endothelial cells with mesenchymal stem cells [79], an alternative approach could consist of the angiogenic activation of the endothelial cells induced by the microstructures, independently of exogenous VEGF treatment. However, such an approach would necessitate further study of the multicellular organization in derived microstructures, which is beyond the scope of this work. Second, the microstructures could be used for screening approaches in the field of VEGF-independent angiogenesis events. Indeed, tumour therapies based on anti-angiogenic (anti-VEGF) drugs often fail because of escaping and resistance processes, involving the emergence of alternative VEGF-independent pathways. Such pathways might partly be governed by geometrical cues and mechanical constraints in the remodelled tumour microenvironment. VEGF-independent tip-like phenotypes observed in our microstructures could constitute a valuable tool to screen drugs targeting these resistant pathways, in the context of a completely controllable system with easy visualization and advanced tools for automated quantification.

## 4. Materials and Methods

### 4.1. Two-Photon Polymerization

The two-photon polymerization set-up previously described [37] consisted of a QSwitch Teem Photonics laser (Grenoble, France), 10 kHz, 5 ns pulses, 10 µJ, 532 nm, a IX70 microscope with a water objective 60× (NA 1.2) LPlanApo, Olympus, a piezo-z stage and a 3D stage (Physik Instrumente, Karlsruhe, Germany), and a Guppy CCD camera for monitoring structure formation. It was driven by Lithos software, with an autofocus module [80]. Norland optical adhesive NOA61 (Norland Products, Cranbury, NJ, USA) was used directly on bare glass coverslips for two-photon polymerization. After deposition on a 30 mm silanized or bare circular coverslip of one drop of NOA61 resin, the initial z position of the sample was adjusted so that the focal volume was just above the glass surface, and the microstructure was polymerized by moving the focal volume. Typical parameters used were: laser power at the objective front aperture, 0.5–0.8 mW; exposure time, 6–8 ms. After completion, structures were washed sequentially with acetone and ethanol. The two-photon microfabricated scaffolds were observed using a Scanning Electron Microscope (FEG-SEM LEO 1530, LEO Elektronenskopie GmbH, Oberkochen, Germany) after the samples were gold sputtered. Alternatively, the autofluorescence of microstructures allowed us to visualize them by confocal imaging. Structure autofluorescence was reduced by coating with Sudan Black B (SBB, Sigma Aldrich, Merck, Burlington, MA, USA) for time-lapse experiments. After the washing steps, structures were incubated in SBB at a concentration of 0.1% *w*/*v* in 70% ethanol for 1 h. The sample was then rinsed several times with 100% ethanol.

### 4.2. Cell Culture and Treatments

HUVECs were either from a commercial source with pooled donors or from home-made prepared primary cultures, with similar behaviours. HUVECs used for experiments with labelling on fixed cells were from Lonza, Basel, Switzerland (C2519A, human umbilical vein endothelial cells, pooled). These experiments were realized from three different stocks (amplified in Lonza EGM-2 medium and frozen at their first passage). HUVECs used in live experiments were labelled directly from primary cultures. They were prepared from human umbilical cords provided by AP-HP, Hôpital Saint-Louis, Unité de Thérapie Cellulaire, CRB-Banque de Sang de Cordon, Paris, France. HUVECs Lifeact-GFP cells were obtained from these cells by transduction with rLVUbi–LifeAct^®^–TagGFP2 (Ibidi, Gräfelfing, Germany). After thawing, cells were maintained for 1–2 passages in endothelial cell growth medium 2 (ECGM2, Promocell GmbH, Heidelberg, Germany), at 37 °C and 5% CO_2_, in collagen-coated flasks. Prior to cell seeding, microstructures were first sterilized in ethanol and rinsed three times with PBS. No additional coating was performed, since initial tests with collagen coating of the microstructures did not lead to clear differences in cell coverage compared with uncoated structures. Cells were seeded at 20,000–30,000 cells/cm^2^ in ECGM2 supplemented with penicillin–streptomycin. For some experiments without VEGF, either endothelial cell growth medium (ECGM, Promocell GmbH, Heidelberg, Germany) or ECGM2 without VEGF were used. Cultures were maintained for 1–5 days before fixation; the duration of culture on structures was adjusted with the criteria of having ~1 day of cell coverage on structures before fixation as observed by visual monitoring, and was dependent on the initial local cell surface density. For VEGF treatment, vascular endothelial growth factor (VEGF165, Thermo Fisher Scientific, Waltham, MA, USA) was added a few hours after seeding at different concentrations (see main text), and the medium with VEGF changed every day, including a few hours before fixation. For the assessment of VEGFR2 phosphorylation upon VEGF treatment, cells on microstructures (day 2) were depleted in growth factors for 4 h (endothelial cell basal medium, + 1% bovine serum albumin), incubated 10 min in the same media with or without VEGF (50 ng/mL), and immediately fixed. For pharmacological treatments used in living experiments, first, a movie was realized before drug addition, and second, a movie in the same conditions immediately after drug addition. It was independently checked that DMSO addition at the maximal concentration coming from drug addition (1/1000) did not alter the dynamic behaviour. ML-7 (hexahydro-1-[(5-iodo-1-naphthalenyl)sulfonyl]-1H-1,4-diazepine, Bertin Bioreagent, Montigny le Bretonneux, France), Y-27632 (*4-[(1R)-1-aminoethyl]-N-4-pyridinyl-trans-cyclohexanecarboxamide*, Bertin Bioreagent, Montigny le Bretonneux, France) and FAK inhibitor PF-573228 (*6-[4-(3-Methanesulfonyl-benzylamino)-5-trifluoromethyl-pyrimidin-2-ylamino]-3,4-dihydro-1H-quinolin-2-one*, Sigma Aldrich, Merck, Burlington, MA, USA) were used.

### 4.3. Cell Labelling

Cells on structures were washed three times with 37 °C PBS+ (phosphate-buffered saline +0.1 mM CaCl_2_ + 0.1 mM MgCl_2_), fixed with 4% paraformaldehyde in PBS^+^ (16% PFA, Electron Microscopy Sciences, diluted in PBS^+^) and permeabilized with 0.1% Triton-X100. Cells were basically labelled with 1 µg/mL Hoechst 34580 and with 1 µg/mL phalloidin-TRITC (Sigma). For some experiments, membrane was labelled with fluorescent lectins (wheat germ agglutinin, Sigma, 10 µg/mL), or indirect immunofluorescence was performed. For VE-cadherin labelling, an anti-VE-cadherin antibody (Abcam, Cambridge, UK; ab 33168, dilution 1:200) and a secondary anti-rabbit antibody, IgG coupled to Alexa647 (Invitrogen, ThermoFisher Scientific, Waltham, MA, USA, dilution 1:1000) were used. For phosphorylated VEGFR2 labelling, an anti-phosphorylated VEGFR2 (Tyr1175) antibody from Cell Signalling (19A10, dilution 1:200) and a secondary antibody, anti-rabbit IgG coupled to Alexa488 (Invitrogen, ThermoFisher Scientific, Waltham, USA, dilution 1:1000) were used. After labelling, structures with cells were kept in PBS^+^ and imaging was performed in PBS^+^ without mounting medium.

### 4.4. Imaging and Image Analysis

Image acquisitions were performed using: (1) for fixed samples, confocal TCS SP_CSU, with resonant scanner (12 kHz), 63X objective, with *z* compensation (Imaging Facility of Institute Pierre-Gilles de Gennes, Paris, France), and (2) for experiments on living cells, spinning disks from the PICT-IBiSA platform, equipped either with an inverted Eclipse Ti-E (Nikon, Tokyo, Japan) and a spinning disk CSU-X1 (Yokogawa) integrated in Metamorph software by Gataca Systems, with a Prime BSI or QuantEM camera (Photometrics, Tucson, AL, USA) and a *z*-motor Nanoz100 (Mad City Lab, Madison, WI, USA), temperature and CO_2_ controllers from Life Imaging Systems, and with a CFI Plan Apo 60X Oil (NA 1.4) objective.

ImageJ was used for rotation, color merging and contrast enhancement. Brightness and contrast were optimized for visual representation. Partial maximal *z* projections are shown for “bottom” representation (contact with glass substrate), middle (closed part of the open structures), “below” (1.2 µm below the top of the structures), and top (above the structures). *z* projections were performed on 2–4 confocal planes acquired with 0.3 µm *z* spacing for bottom and middle/below views, and 4–8 planes (about 2 µm high) for top views. In some cases, denoising was performed using the Safir filter ImageJ plugin [81]. Three-dimensional visualization was performed with the ImageJ 3DViewer plugin (https://imagej.net/3D_Viewer, accessed 1 March 2020) [82]. Alternatively, median filter and background subtraction were used for image presentation from time-lapse experiments. For some images, reconstruction of the whole structure from the different fields was performed with the ImageJ Stitching plugin.

Home-made software was specifically developed for 3D automated detection of nuclei, semi-automatic filopodia detection in fixed samples and automatic filopodia detection and tracking in living experiments in our structures, as detailed in the Appendix B (Figure A1 and Figure A2). Briefly, filopodia were detected using a Convolutional Neural Network (CNN) model trained with semi-automatically labelled images from fixed experiments. The detection map provided by the CNN was then segmented and the resulting objects were tracked along time, with linear assignment performed according to an Intersection over Union (IoU) metric. The barbed (+) and pointed (−) extremities of filopodia were computed based on their distance to the cell body.

The error bars shown correspond to standard deviation. Statistical tests were performed on scipy.stats: paired *t*-tests were performed after checking the normality of the distributions of the difference between paired conditions with a Shapiro–Wilk test. The violin plots (Figure 5) represent the data distribution and are based on a Gaussian kernel density estimation using the Scott’s rule as implemented in matplotlib. For the representation of filopodia length distributions in Figure 9, Matlab (Mathworks, Natick, Massachussetts, USA) violin function was used, with bandwidth 0.5 (Hoffmann H, 2015: violin.m—Simple violin plot using Matlab default kernel density estimation. INRES (University of Bonn), hhoffmann@uni-bonn.de). For smoothing the kinetic curves, a Savitzky–Golay filter was implemented from scipy.signal library, with a window size of 21 time intervals (7 min) and an interpolation with a third-degree polynomial. For the determination of filopodia lifetime, the filopodia that are already present at the beginning or at the end of a movie introduce a bias in the lifetime calculations, since their lifetime cannot be known. In order to remove as much as possible of this effect, first, we did not consider in our interpretation a time interval of a few minutes at the beginning or at the end of the movie (duration corresponding to filopodia lifetime; see dotted areas in the lifetime plots), and second, we discarded all filopodia that we did not see appear and disappear during the movie. For Figure 6 and Table 1, means were performed from intervals of 16.6 min per movie, systematically excluding the beginning and the end of the movies in order to reduce the bias in the estimation of filopodia lifetimes. For Figure 8 and Figure 9, means were performed on time intervals with a constant duration of 10.3 min, defined as follows: for normalized numbers, Ctr (before drug addition), interval immediately before drug addition, and drug, interval beginning 21.3 min after drug addition. For lifetimes, since the 5 last minutes before drug addition could not be considered (see above), the Ctr sampling interval was chosen to end 5 min before drug addition. The Drug treatment was begun 21.3 min after drug addition, except for PF-573228 where the effects fully developed at a longer time (beginning 40 min after drug addition).

## Figures and Tables

**Figure 2 ijms-23-02415-f002:**
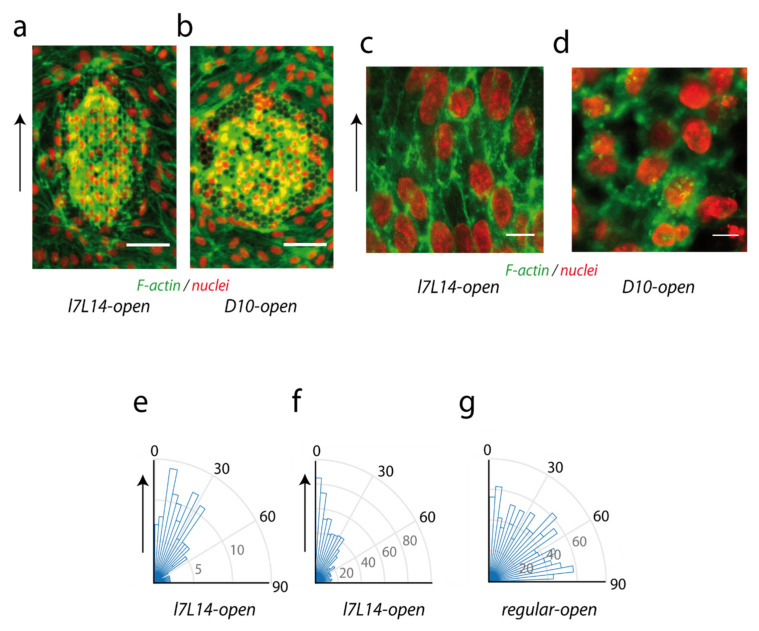
Cellular organization in the top monolayer. (**a**,**b**) Global HUVECs coverage of (**a**) *l7L14-open* and (**b**) *D10-open* microstructures. VEGF 0.5 ng/mL. *z* projection is shown, red: nuclei, green: F-actin, bar 50 µm. (**c**,**d**) Detail of the top monolayer covering (**c**) *l7L14-open*, with the axis of elongation of hexagons represented by a black arrow (top, left), and (**d**) regular *D10-open* microstructures. Red: nuclei, green: F-actin, *z* projection of planes above structures, cells fixed at time of structure colonization (2 (**b**,**d**) or 3 (**a**,**c**) days after cell seeding). Scale bar 10 µm. (**e**–**g**) Histograms of nuclei orientations in the top layer above the microstructures. Only the part of the nuclei above 4 µm height was considered. (**e**) *l7L14-closed* (127 nuclei in 7 structures) (**f**) *l7L14-open* (700 nuclei in 15 structures) (**g**) *D4.5*, *D6.6*, *D8.8*, *D10*, and *D13.5-open* microstructures (1051 nuclei in 23 structures). Vertical arrows in (**a**,**c**,**e**,**f**) represent the reference axis for elongated structures. For regular structures, the reference axis was chosen perpendicular to one size of the hexagon (not shown).

**Figure 3 ijms-23-02415-f003:**
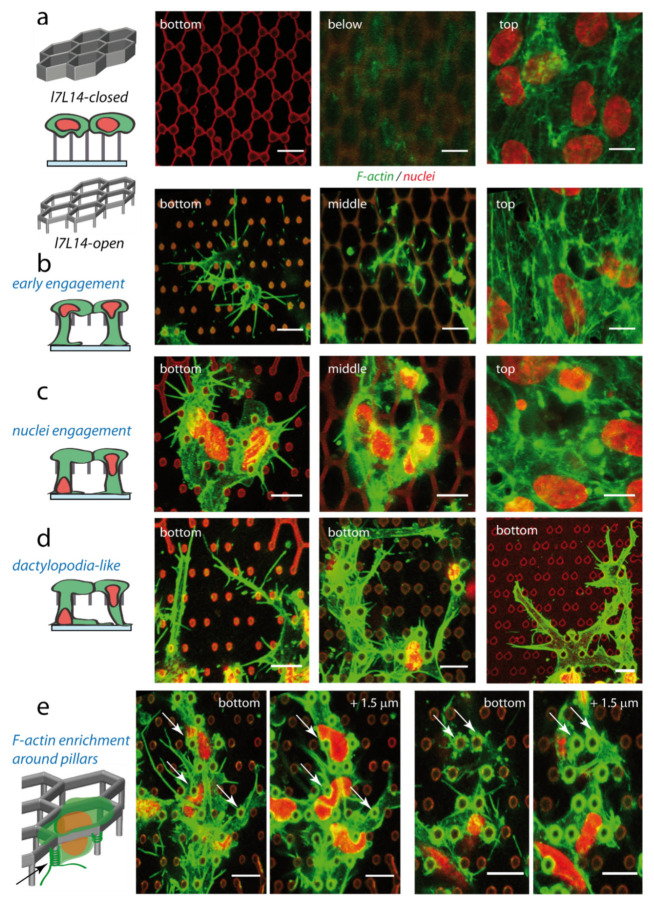
HUVECs organization on closed and open microstructures. Left, schemes of hexagonal lattices, and side or 3D views of typical HUVECs organization after colonization. Right, confocal planes, denoised images, scale bar 10 µm. Red: nuclei and microstructure autofluorescence, green: F-actin. Cells were cultivated in standard conditions (VEGF 0.5 ng/mL) unless otherwise specified. Cells were fixed at time of structure colonization (2 (**b**,**d**-middle, **d**-right), 3 (**d**-left) or 4 (**a**,**c**) days after cell seeding). (**a**) Closed microstructures. Confocal planes on HUVECs on *l7L14-closed* microstructure. Left, bottom plane (contact with the bottom substrate); middle, just below the structure (1.2 µm below the top of the structure); top, above the structure. (**b**–**e**) Typical examples of colonization of open *l7L14-open* structures, with vertical engagement and formation of filopodia in the bottom plane: (**b**) with limited cell area on the basal substrate), (**c**) with larger membrane and nuclei vertical engagement, (**d**) with formation of extended, dactylopodia-like protrusions, and (**e**) typical enrichment observed for F-actin around the micropillars, white arrows. (**b**,**c**) From left to right, bottom plane; middle plane (at the level of the closed part of the structure); top, above the structure. (**d**) Dactylopodia-like protrusions were observed without (**d**-left) or with (**d**-middle,right) VEGF in the media. Bottom planes are shown. For **d**-left, correspondent membrane labelling with bottom, middle and top views is shown in Figure A3c. (**e**) Two examples of bottom and top planes are shown.

**Figure 4 ijms-23-02415-f004:**
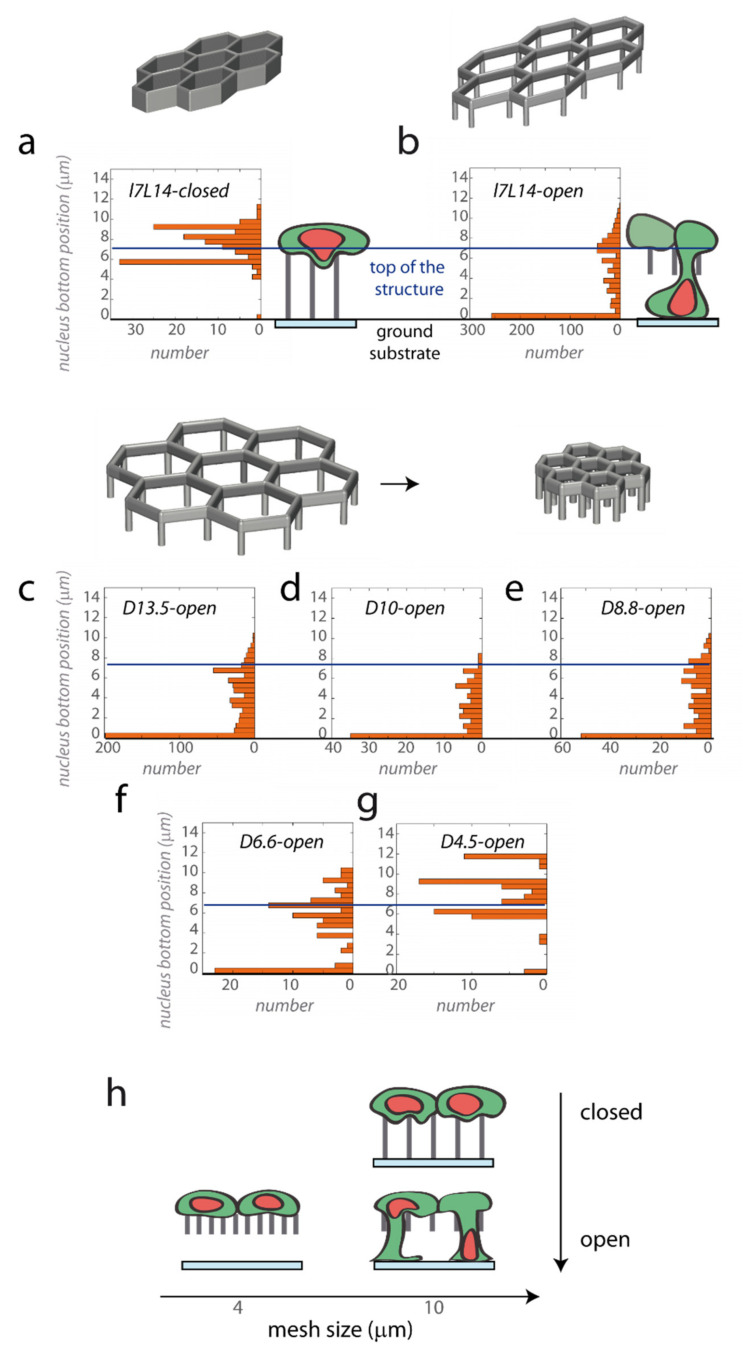
Dependence of geometry and pore size for nuclei engagement. HUVECs were cultivated on elongated (**a**) *l7L14-closed* microstructures (127 nuclei in 7 structures) and (**b**) *l7L14-open* structures (700 nuclei in 15 structures), or on regular open structures with different horizontal dimensions *D*: (**c**) *D13.5-open* (607 nuclei in 6 structures), (**d**) *D10-open* (95 nuclei in 1 structure), (**e**) *D8.8-open*, (178 nuclei in 4 structures), (**f**) *D6.5-open* (94 nuclei in 5 structures), and (**g**) *D4.5-open* (77 nuclei in 7 structures). After the 3D segmentation of nuclei, the position of the bottom of the nuclei was determined, with a position of 0 µm corresponding to the contact with the bottom substrate (black horizontal line) and a position of 7 µm to the top of the microstructures (blue horizontal line). For *D4.5-open* microstructures (**g**), the highest position compared with the expected height of the structure likely reflects an optical deformation when imaging through a densely polymerized structure. (**h**) Recapitulative scheme showing nuclei engagement in the function of the pore size and on the closed or open nature of the microstructure.

**Figure 5 ijms-23-02415-f005:**
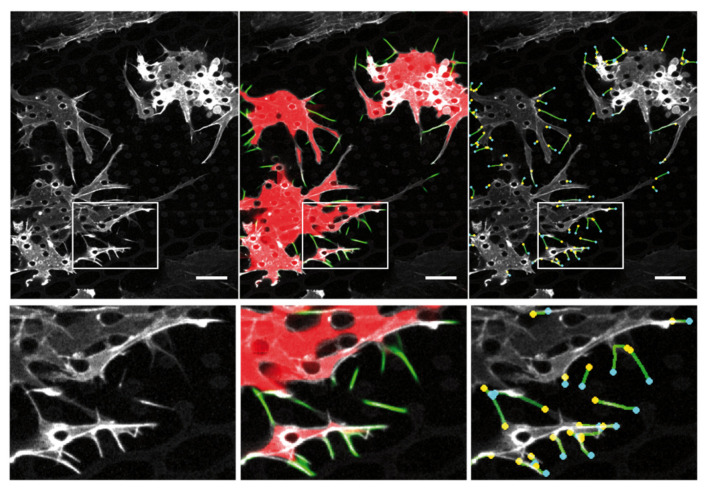
Automated filopodia detection. Left, original image of Lifeact-GFP HUVECs cells in the microstructures. Middle, automatic detection of cell islets (**red**) and filopodia (**green**). Right, detection of (−) and (+) extremities for each individual filopodia, used for filopodia tracking. Bottom, zoom on the image part in the white rectangle on top. Scale bar 15 µm.

**Figure 6 ijms-23-02415-f006:**
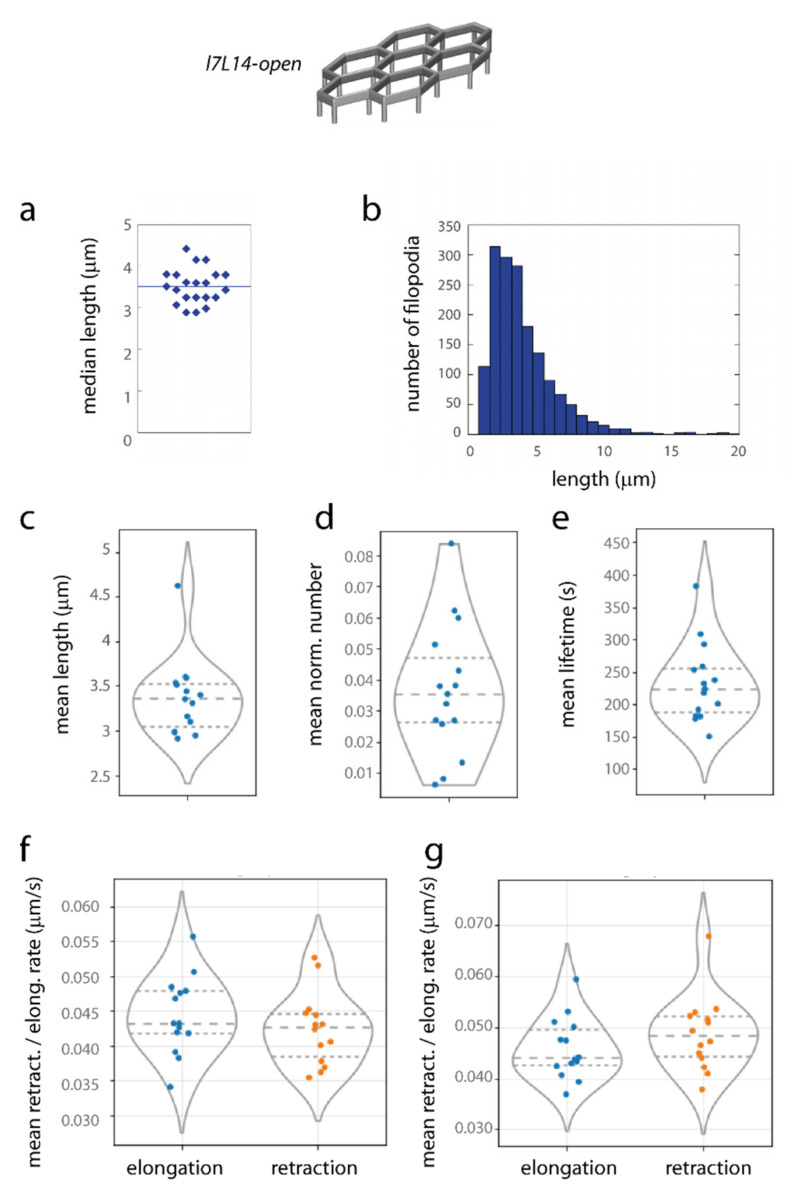
Quantification of filopodia organization. HUVECs were cultivated on *l7L14* microstructures, and filopodia quantified. (**a**) Median filopodia lengths from analyses on fixed samples, one point corresponds to one structure (bar: average value). (**b**) Histogram of filopodia lengths from analyses on fixed samples. (**c**–**g**) Mean values from analyses on timelapse images. Each point of the plot corresponds to the mean filopodia values on one movie (*n* = 15 independent movies were analysed, with a typical number of 10–100 filopodia detected at a given time point for each movie). Violin plots are shown, with dotted lines representing quartiles (middle line: median). (**c**) filopodia length, (**d**) number of filopodia normalized by the perimeter (in µm) of the cell islets inside the structure, (**e**) filopodia lifetime, (**f**,**g**) elongation (blue) and retraction (orange) rates of the (−) (**f**) and (+) (**g**) filopodia extremities.

**Figure 7 ijms-23-02415-f007:**
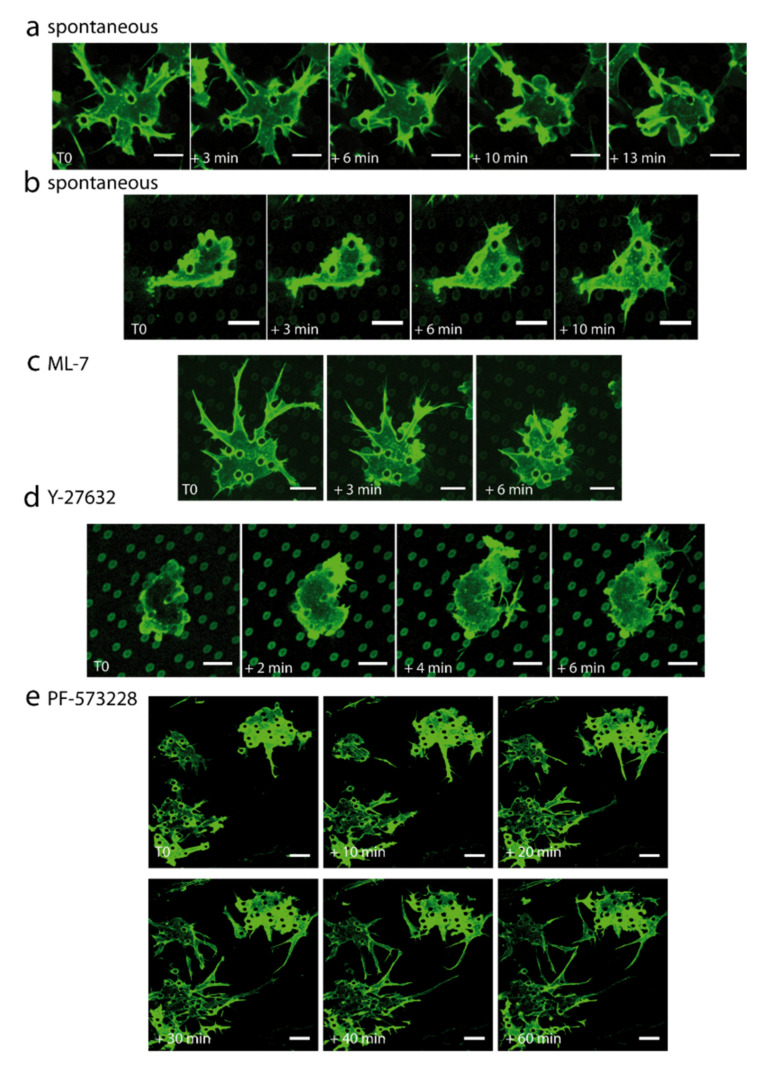
Transition between different protrusive modes in the microstructures, and molecular mechanisms. (**a**,**b**) Spontaneous transitions between an amoeboid state and a mesenchymal state (**a**), and between a mesenchymal and an amoeboid state (**b**). (**c**–**e**) Kinetics of the representative effects of (**c**) ML-7 (10 µM), (**d**) Y-27632 (10 µM), or (**e**) PF-573228 (10 µM) addition on Lifeact-GFP HUVECs cells (green, Lifeact-GFP), *z* projection of the bottom planes. (**a**,**b**) T0 refers to an arbitrary moment chosen as time origin, (**c**–**e**) T0 refers to the time of drug addition. Movies were acquired 1 day after cell seeding. Scale bar 10 µm (**a**–**d**), 15 µm (**e**).

**Figure 8 ijms-23-02415-f008:**
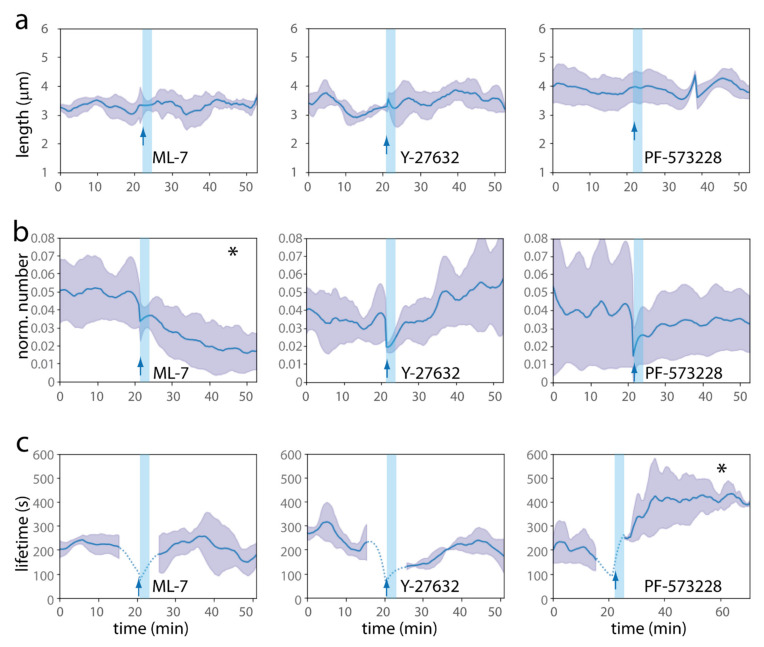
Quantitative analysis of ML-7, Y-27632 and PF-573228 effect on filopodia dynamics. (**a**–**c**) Evolution along time of (**a**) mean filopodia lengths, (**b**) normalized number of filopodia (normalization by the perimeter, in µm, of the cell islets inside the structure), and (**c**) mean lifetimes of filopodia present at this time, for ML-7 (left), Y-27632 (middle) and PF-573228 (right) treatments, *n* = 3 independent experiments per condition. Error: S.D. Pre-treatment and post-treatment graphs are concatenated. The blue arrow indicates the time of drug addition. The light blue vertical bar indicates a state of stabilization of the system after media change (~3 min long). The dotted lines for lifetime (**c**) refer to times where the lifetime cannot be precisely assessed, because of the imaging interruption during media changes (durations, ~1 mean filopodia lifetime before and after drug addition; see Materials and Methods). * in (**b**) ML7, and (**c**) PF-573228, indicates statistically significant differences before and after drug addition, with *p* < 0.05 (see Figure A7b–d).

**Figure 9 ijms-23-02415-f009:**
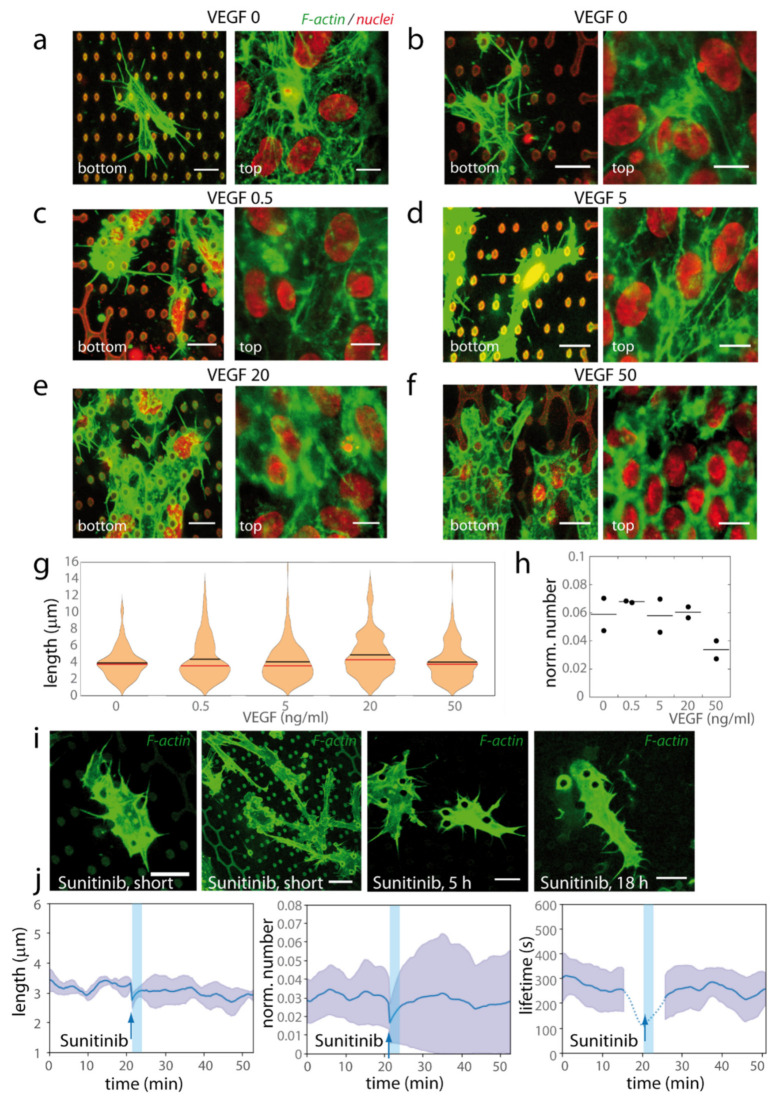
Dependence on VEGF. (**a**–**h**) HUVECs were cultivated on *l7L14-open* microstructures, and long-term cultures were performed in different VEGF concentrations. Cell organization was studied from fixed samples. (**a**,**b**) Culture without VEGF: (**a**) ECGM media (Promocell), which contains no VEGF supplement, (**b**) ECGM2 media without the VEGF supplement. (**b**–**e**) Standard ECGM2 media with (**c**) 0.5 ng/mL VEGF, (**d**) 5 ng/mL, (**e**) 20 ng/mL, and (**f**) 50 ng/mL. (**a**–**f**) Red, nuclei (and structure autofluorescence), green, F-actin. Bottom and top planes are represented. Cells fixed at time of structure colonization (2 (**b**–**f**) or 3 (**a**) days after cell seeding). Denoised images, bar 10 µm. (**g**,**h**) Quantification for experiments realized in parallel in the different VEGF concentrations (2 days after cell seeding). The figure corresponds to an experiment where the five VEGF concentrations were analysed simultaneously, with two structures per condition. The generation of numerous filopodia for the conditions VEGF0 was observed in 3 additional independent microstructures (see raw data). (**g**) Histograms of filopodia lengths (black bar, mean value, red bar, median value). (**h**) Number of filopodia normalized by the perimeter on cell bodies in the bottom part of the structure. One point corresponds to one structure. (**i**,**j**) Effect of sunitinib (300 nM) treatment on HUVECs-Lifeact-GFP cells. Movies were acquired 1 day after cell seeding. (**i**) After sunitinib addition, cells still exhibited filopodia (**left**) and dactylopodia-like protrusions with filopodia (**middle-left**) (26 and 29 min after drug addition, respectively). Dynamic filopodia were still present after longer sunitinib treatments: 5 h (**middle-right**) or 18 h (**right**). Green, Lifeact-GFP, *z* projection of the bottom planes. Scale bars 10 µm (**left**, **middle-right**, **right**), 15 µm (**middle-left**). (**j**) Evolution along time of the mean filopodia length (**left**), of the number of filopodia normalized by the perimeter (in µm) of the cell islets inside the structure (**middle**), and of the mean lifetime of filopodia present at this time, upon sunitinib treatment, *n* = 3 experiments per condition, errors: S.D. (*n* = 3 independent experiments). Pre-treatment and post-treatment graphs are concatenated. The blue arrow indicates the time of drug addition. The light blue vertical bar indicates a state of stabilization of the system after media change (~3 min long). The dotted lines for lifetime (**c**) refer to times where the lifetime cannot be precisely assessed, because of the imaging interruption during media changes (durations, ~1 mean filopodia lifetime before and after drug addition; see Materials and Methods). No statistical difference was detected in filopodia dynamics upon sunitinib treatment (see Figure A7e).

**Figure 10 ijms-23-02415-f010:**
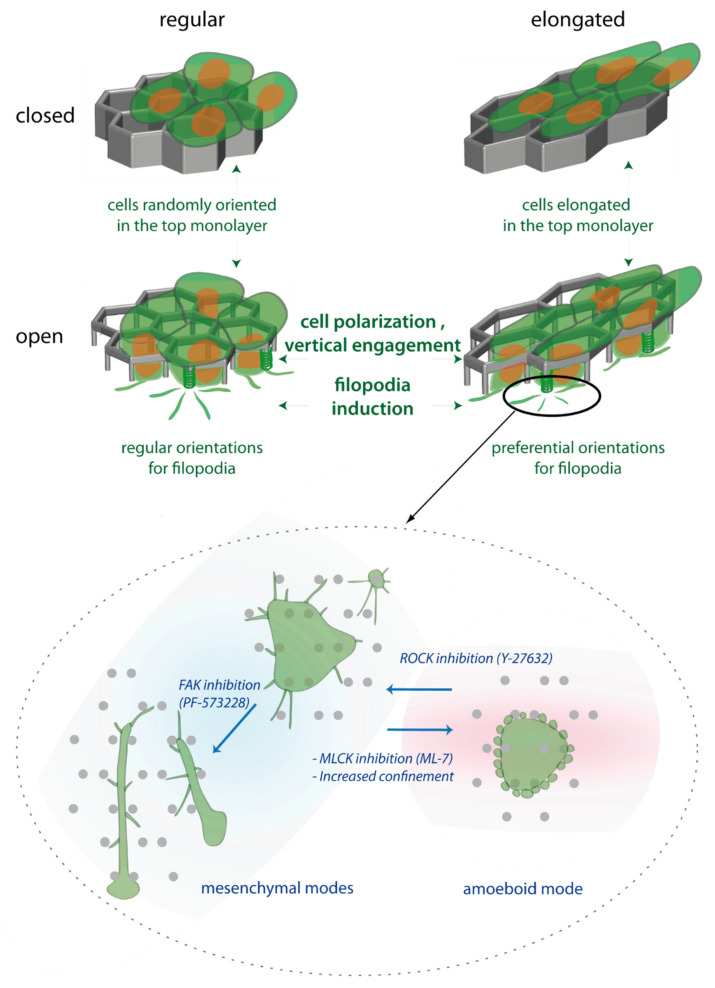
Recapitulative scheme of the global cell behaviour in the microstructures. (**Top**) Organisation of endothelial HUVECs in function of the elongation of hexagons in the horizontal plane, and of the open or closed nature of the microstructure. These two characteristics modulate the orientation of the top monolayer, the vertical engagement and the induction of filopodia, and the orientation of the exploratory filopodia. The vertical cell polarization and the induction of filopodia are hallmarks of tip cell phenotype, but occur independently of VEGF. (**Bottom**) Zoom of the different protrusive modes present in the bottom plane (pillars) of open elongated structures, and transitions between them.

**Table 1 ijms-23-02415-t001:** Filopodia characteristics in the microstructures. Values extracted from timelapse movies (*n* = 15, error: S.D.).

Quantity	Extremity	Value
Length (μm)		
		3.36 ± 0.41

Lifetime (s)		
		233 ± 58
Elongation speed (nm/s)	(+)-end	45.8 ± 5.8
	(−)-end	44.3 ± 5.4
Retraction speed (nm/s)	(+)-end (−)-end	48.6 ± 7.0
		42.4 ± 5.0
Normalized number (μm^−1^)		0.037 ± 0.020

## Data Availability

Both the raw data and processed data required to reproduce these findings are available to download from Zenodo, at https://zenodo.org/record/6198946 accessed on 20 December 2021.

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
