# Peer review of "Dynamics of Endothelial Engagement and Filopodia Formation in Complex 3D Microscaffolds"

_ijms, 2022, doi:10.3390/ijms23052415_

Round 1
Reviewer 1 Report
- MLCK in the abstract, please add complete words (myosin light chain kinase) plus abbreviations. line 27
- How do you suggest that these behaviors are just induced by microtopography? Gene expression and signaling pathway analysis also VEGF (internal) protein detection are not considered in this experiment. Line 85-86.
- L7L14-open, with the axis of elongation of hexagons represented by a yellow arrow (top, left), where is the yellow arrow? There is a black arrow left side of the figure. Line 153
- Culturing time was not mentioned by the authors on figures information.
- Please check the punctuation marks. Line 248- 251
- Figure 5. Please add scale bars. Line 281.
- Figure 7. Please add scale bar on all figures.
- In figure 7 a-b, no drugs were added for spontaneous condition. So why did authors add TO on this figures? Line 381
- Figure 7, part b, first figure, please consider scale bar on the same side with other figures.
- Some references suggest several hours for treatment by Sunitinib but in this experiment, the authors treat the cells for a few minutes. Please explain why you choose the short time and how you expect this time is enough?
Reviewer 2 Report
In this manuscript, Ucla and colleagues demonstrate the biophysical parameters experienced by endothelial cells when cultured on open or closed bioengineered substrates configuration. The authors nicely demonstrate the formation of protrusions (identified as filopodia or dactylopodia) under these engineered settings and the orientation of these structures are regulated by myosin light chain (MLCs) and ROCK signaling and Focal adhesion kinases. These studies represent the biomechanical aspects of filopoidia generated by endothelial tip cells necessary for angiogenesis. In general, the paper is sound with several key observations and automatic quantification of protrusive EC structures are of interest to the field. I would suggest the authors undertake the following suggestions to improve the physiological relevance of their findings and thereby make their study robust for publication.
Specific comments
1) The rationale for building and transition between the open and closed configurations of ECs is not clear across the manuscript!
2) What does the elongation of ECs under Fig a1d-h? Are these referring to fluid shear stress experienced by ECs under blood flow? Are these conditions (as generated by the authors) more angiogenic? How do we experimentally assess this? At least the authors should shed some experimental insights on these parameters. The authors should provide some tube-formation experiments with these cells that have been placed on open or closed conformation.
3) An experiment comparing the migration of HUVECs in open or closed states (Fig 3) would be of value! Live imaging of these cells will shed light on the cellular migratory speeds.
4) The automated quantification of flipodia in the manuscript is nice and the authors are commended for doing a generic good job! However, it is not surprising that MLCk or ROCK will be inhibiting these structures as multiple previous reports have illustrated their importance for the regulation of protrusive cell structures.
5) The authors suggest that RTK inhibition does not necessarily affect the filopodia in their settings in Fig 9? Why is that so? Since VEGFR2 is a key regulator of angiogenesis and tip cell migration, how do they align with this finding to the broader physiological relevance?
Round 2
Reviewer 1 Report
The revised version looks ok to me.
Reviewer 2 Report
The authors have done a good job in responding to my previous comments. Their explanation and new additions have improved the manuscript. Hence, I am now recommending the manuscript for publication.